# Intraplaque Neovascularization, CD68+ and iNOS2+ Macrophage Infiltrate Intensity Are Associated with Atherothrombosis and Intraplaque Hemorrhage in Severe Carotid Atherosclerosis

**DOI:** 10.3390/biomedicines11123275

**Published:** 2023-12-11

**Authors:** Ioan Alexandru Balmos, Mark Slevin, Klara Brinzaniuc, Adrian Vasile Muresan, Horatiu Suciu, Gyopár Beáta Molnár, Adriana Mocian, Béla Szabó, Előd Ernő Nagy, Emőke Horváth

**Affiliations:** 1Doctoral School of Medicine and Pharmacy, I.O.S.U.D., George Emil Palade University of Medicine, Pharmacy, Science, and Technology of Targu Mures, 540142 Targu Mures, Romania; ioan.balmos@umfst.ro (I.A.B.); gyoparbeata@yahoo.com (G.B.M.); scarlat.adriana05@gmail.com (A.M.); 2Department of Anatomy, George Emil Palade University of Medicine, Pharmacy, Science, and Technology of Targu Mures, 540142 Targu Mures, Romania; 3Vascular Surgery Clinic, County Emergency Clinical Hospital of Targu Mures, 540136 Targu Mures, Romania; 4Center for Advanced Medical and Pharmaceutical Research (CCAMF), George Emil Palade University of Medicine, Pharmacy, Science and Technology, 540142 Targu Mures, Romania; mark.slevin@umfst.ro; 5M3 Department of Surgery, George Emil Palade University of Medicine, Pharmacy, Science, and Technology of Targu Mures, 540142 Targu Mures, Romaniabela.szabo@umfst.ro (B.S.); 6Emergency Institute for Cardiovascular Diseases and Transplantation, 540142 Targu Mures, Romania; 7Pathology Service, County Emergency Clinical Hospital of Targu Mures, 50 Gheorghe Marinescu Street, 540136 Targu Mures, Romania; emoke.horvath@umfst.ro; 8Department of Biochemistry and Environmental Chemistry, George Emil Palade University of Medicine, Pharmacy, Sciences, and Technology of Targu Mures, 540142 Targu Mures, Romania; 9Laboratory of Medical Analysis, Clinical County Hospital Mures, 540394 Targu Mures, Romania; 10Department of Pathology, Faculty of Medicine, George Emil Palade University of Medicine, Pharmacy, Science, and Technology of Targu Mures, 38 Gheorghe Marinescu Street, 540142 Targu Mures, Romania

**Keywords:** carotid atherosclerosis, macrophage, intraplaque neovascularization, atherothrombosis, intraplaque hemorrhage

## Abstract

Background: Atherosclerosis is a progressive disease that results from endothelial dysfunction, inflammatory arterial wall disorder and the formation of the atheromatous plaque. This results in carotid artery stenosis and is responsible for atherothrombotic stroke and ischemic injury. Low-grade plaque inflammation determines biological stability and lesion progression. Methods: Sixty-seven cases with active perilesional inflammatory cell infiltrate were selected from a larger cohort of patients undergoing carotid endarterectomy. CD68+, iNOS2+ and Arg1+ macrophages and CD31+ endothelial cells were quantified around the atheroma lipid core using digital morphometry, and expression levels were correlated with determinants of instability: ulceration, thrombosis, plaque hemorrhage, calcification patterns and neovessel formation. Results: Patients with intraplaque hemorrhage had greater CD68+ macrophage infiltration (*p* = 0.003). In 12 cases where iNOS2 predominated over Arg1 positivity, the occurrence of atherothrombotic events was significantly more frequent (*p* = 0.046). CD31 expression, representing neovessel formation, correlated positively with atherothrombosis (*p* = 0.020). Conclusions: Intraplaque hemorrhage is often described against the background of an intense inflammatory cell infiltrate. Atherothrombosis is associated with the presence of neovessels and pro-inflammatory macrophages expressing iNOS2. Modulating macrophage polarization may be a successful therapeutic approach to prevent plaque destabilization.

## 1. Introduction

Carotid atherosclerotic disease is a complex condition characterized by endothelial dysfunction, damage and progressive inflammatory changes, which can be a precursor to stroke, leading to irreversible neurological deficits or death. Stroke, a pathology of the elderly, is the second leading cause of death worldwide according to the latest statistics and may appear in the case of moderate and severe carotid atherosclerosis, usually symptomatic, due to unstable plaques [1,2,3]. Severe stenosis caused by atherosclerosis of the carotid artery is responsible for more than 15% of strokes and transient ischemic attacks (TIAs), and this stenosis occurs at the level of the common and internal carotid arteries [4,5].

Clinical studies have demonstrated that the grade of the stenosis is poorly correlated with the symptomatology; therefore, intricate mechanisms might determine the activity of a carotid lesion. Histopathological analysis of fragments removed by vascular surgery showed a close correlation between plaque morphology and neurological symptoms (transient ischemic attack, minor stroke and major stroke). Patients with “vulnerable plaques” (with surface ulceration, intraplaque hemorrhage, thinner fibrous cap and rich neovascularization) are prime candidates for stroke because of a high risk of embolization and thrombosis [6]. These vulnerable carotid plaques are characterized by a chronic inflammatory process that develops in the endothelial layer of the arterial wall, involving the influx of the monocytes from the blood, activation and migration of macrophages, development of lipid deposits, smooth muscle proliferation and the appearance of neovascularization [7,8]. Lymphocytes also infiltrate the plaque and contribute to low-grade local inflammation.

There is a cause–effect relationship between arterial wall inflammation and neovascularization of the plaque promoted by the release of vascular endothelial growth factor (VEGF) by smooth muscle cells. These immature vessels serve as a port of entry for other inflammatory cells, lipids and even red blood cells that contribute to plaque growth. At the same time, infiltrated macrophages secrete metalloproteinases such as MMP-2, MMP-9 and other collagenases that destroy connective fibrous tissue, thereby stimulating neovascular growth—changes that lead to a vulnerable atherosclerotic plaque [9].

According to the classical viewpoint, morphological and immunohistochemistry analyses of the atherosclerotic plaques identified two types of macrophages in the same plaque—M1 pro-inflammatory macrophages with a glycolytic metabolism, which are dominant in the symptomatic tissues and specific for unstable plaques, and M2 anti-inflammatory macrophages with an oxidative metabolism, which are dominant in the asymptomatic tissues and specific for more stable plaques. Several years ago, based on transcriptomic data, a multi-dimensional model of macrophage activation was elaborated, and now the dichotomic division to M1 and M2 appears to be mechanistic and simplistic [10]. Many marker molecules were associated with these functions, and it has been recognized that the M1 subtype predominantly expresses CD16/32, CD80, CD86, MHC II and iNOS2, whereas the M2 shows more CD163, CD204 (mannose receptor) and Arg1 positivity [11]. The predominance of one of these subtypes is directly influenced by the immune system cells, especially by the cytokines and growth factors [12,13,14,15]. iNOS transforms L-arginine into citrulline and vasodilatory nitric oxide (NO) but also produces free radicals. Arginase acts on the same substrate, degrading it to L-ornithine, but it has two isoforms. Both reduce the bioavailability of arginine for iNOS, but Arg2 causes NOS uncoupling and the production of excess superoxide anions [16]. It is also possible that the arginases diminish the inflammatory vasodilation due to the decrease in NO. Thus, these enzymes, despite the complex phenotypes and activation models of macrophages, may confer a simple, functional classification of macrophages producing ROS (iNOS positivity) and counterparts with rather more reparatory activity (Arg1 positivity).

In addition, immunohistochemistry studies of atherosclerosis identified the presence of neovascularization in the atherosclerotic plaque and indicated that neoangiogenesis plays a role in the progression and complications of plaques [17,18,19,20]. Inflammation and macrophage subtypes have been shown to play an essential role in angiogenesis. Plaques with many neoformation vessels are more unstable and prone to rupture, leading to atherothrombotic complications such as intraplaque hemorrhage [15,21,22,23].

Numerous vascular smooth muscle cells around the atherosclerotic lesion interact with inflammatory populations, providing plaque stability and preventing fibrous cap rupture. The loss of these smooth muscle cells through senescence or apoptosis results in increased macrophage content and necrotic core volume, decreased matrix content, and significant fibrous cap thinning [24,25]. While the grade of carotid artery stenosis is considered by guidelines to be the most important criterion in disease classification and determining the indication for endarterectomy, recent evidence suggests that the characteristics of atherosclerotic plaque may have a more direct influence on the occurrence of stroke than stenosis alone. Therefore, here, we have characterized the heterogeneous inflammatory micro-environment and microvessel composition from a relatively large cohort of endarterectomy specimens, identifying a functional and phenotypical correlation of macrophage subpopulations with the risk of thrombosis [26,27].

## 2. Materials and Methods

### 2.1. Patients and Tissue Fragments

The carotid plaque specimens whose morphological characteristics are processed in this study were obtained via endarterectomy from 119 patients diagnosed with symptomatic carotid artery (CA) stenosis (according to the European Society for Vascular Surgery, 2023 definitions) [26] hospitalized between 2020 and January 2022 at the Vascular Surgery Clinic—County Emergency Clinical Hospital and the Cardiovascular Surgery Clinic—Cardiovascular Disease and Transplant Emergency Institute of Târgu Mureș (Romania). Plaque fragments obtained from the site of maximum stenosis were immediately fixed in 10% buffered formaldehyde, decalcified in ethylene–diamine–tetra-acetic acid (EDTA) solution (pH 7), embedded in paraffin, processed using standard histological methods, and evaluated by histopathological examination.

Based on the results of a previous study [28], 75 cases with an active perilesional inflammatory infiltrate were selected for further immunohistochemical studies to characterize the monocyte–macrophage component of this mononuclear infiltrate.

Of the 75 re-examined sections stained with hematoxylin and eosin (H&E), 67 tissue samples were of sufficient quantity and quality for immunohistochemical examination and digital morphometry. The main morphological changes leading to plaque instability were recorded: new vessel formation (angiogenesis), the pattern of calcification (type, position, and extent), presence and structure of the lipid core (lipid-rich large necrotic core or hyaline-rich core), atherothrombosis, intraplaque hemorrhage, fibrous cap damage (with or without parietal thrombus fragments), each scored as present or absent [28]. We also immunohistochemically characterized the infiltrating macrophages and their subtypes and the density of neovascularization.

Since the type of calcification in fibrohialinous lesions of atheromatous plaque is difficult to distinguish from H&E staining and can be misleading, we also used von Kossa’s special staining to clarify the pattern of calcification, with particular attention to the identification of microcalcification foci.

Atherothrombosis was defined as plaque disruption and consequent platelet deposition on the injured vessel wall [29]. Intraplaque hemorrhage was defined by the extravasation and accumulation of blood components and fibrin deposition within the atheromatous plaque clearly visible on H&E stained sections [30,31]. However, for the selection of cases with this modification, we also used CD31-immunostained sections, which visualize both the density of neovascularization and the rupture of the neovascular wall with consecutive intraplaque hemorrhage. CD31 immunolabelling has also helped to characterize atherothrombosis more accurately, indicating a lack of continuity of the intima at the site of clot adhesion to the inner arterial wall.

### 2.2. Immunohistochemistry for Macrophage Density, Subtyping, and Detection of Vessel Density in the Atherosclerotic Plaque

Anti-CD68 mouse monoclonal antibody, clone IC70A (Agilent, Dako Santa Clara, CA, USA) was used to identify the macrophage population, followed by iNOS2-positive M1 (clone RBT, BioSB) and Arg-1 M2 (clone EP261, BioSB) subtype specification according to the manufacturer’s instructions.

In parallel with the macrophage study, plaque neovascularization was also assessed using an anti-CD31 antibody (clone 1A10, BioSB) combined with an anti-SMA antibody (clone BSB-15, BioSB). EnVision FLEX/HRP (Agilent, Dako Santa Clara, CA, USA) secondary antibody in combination with 3,3’-diaminobenzidine chromogen (DAB) substrate visualized the reaction product as a brown color. Cell nuclei were counterstained with hematoxylin. As a negative control, normal serum was substituted for the primary antibody. We interpreted all three immunomarkers against positive internal controls for these reactions. Positive controls for CD68 and CD31 reactions were immunolabelled foam cells of the lipid core, endothelial cells of the intima and endothelium of the vasa vasorum. SMA expression at the level of immature vessels was reported in the immunolabeling of smooth muscle cells of the media.

### 2.3. Assessment of Intraplaque Neoangiogenesis

The presence or absence of intraplaque neoangiogenesis was initially assessed on H&E-stained sections. Plaques were considered revascularized in the presence of small to large, thin-walled, neoformed vessels with dilated or collapsed lumen without or with poor smooth muscle cells (Figure 1a,b). These vessels covered by CD31-positive endothelium often coexist with CD31-positive endothelial cell buds (vascular precursors) (Figure 1c).

### 2.4. Semi-Quantitative Scoring of the CD68+ Mononuclear Inflammatory Infiltrate

Based on the density of CD68-labelled macrophages around the plaque, they were classified as low-grade (score 1) or high-grade (score 2–3). Score 1 was considered a reduced CD68+ infiltrate, representing less than 5% of the cellular population around the lipid core examined with ob.4 (Figure 2a). Score 2 was characterized by immunolabelled cells between 5 and10% of the total peri-lesional cell pool (Figure 2b). If the number of positive cells exceeded 10%, the case was classified as score 3 (Figure 2c). To accurately characterize the abundance of macrophages, we quantified immunolabeling by determining the positive surface area using a digital morphometry method.

### 2.5. Digital Image Analysis Method to Measure Quantitative Individual Plaque Characteristics (CD68, Arg1, iNOS2, CD31)

A total of 242 microphotographs were obtained from representative regions containing the most cells or vascular elements detected by immunolabelling (hotspot method) at 10× magnification using the AxioLab5 microscope connected to a Zeiss AxioCam 8 digital camera (Figure 3).

Quantitative analysis of all carotid plaque images was performed using ImageJ software (ImageJ 2 for macOS, version 2.3.0, NIH, National Institute of Health, Bethesda, MD, USA). Images were imported into ImageJ software. The percentage of CD68-, iNOS2-, Arg1- and CD31-positive cells was calculated relative to the total area of the imaged area (positive relative area). For the Arg1/iNOS2 composition, we defined a cut-off value of 1 and classified our cases into Arg1-dominant and iNOS2-dominant cases (Figure 4). We considered iNOS2 characteristic for M1 and Arg1 for M2 macrophage subtypes.

### 2.6. Statistical Analysis

Variables with discrete values and transformed continuous variables were assessed for absolute and relative distribution frequencies. We performed the analysis of 2 × 2 contingency tables with Fisher’s exact test and of 3 × 2 contingency tables with the Pearson χ^2^ test. Non-linear logistic regression models were set up to determine the predictors of ulceration and atherothrombosis. In all tests, *p* values < 0.05 were considered statistically significant. Procession and statistical analysis of data were performed using Microsoft Excel 2016 (Microsoft Corporation, Redmond, WA, USA) and GraphPad Prism 9.5.1 (GraphPad Software LLC., San Diego, CA, USA).

## 3. Results

### 3.1. Study Group Characteristics

A total of 67 cases (47 men and 20 women) with a mean age of 65 years were included. All patients had severe carotid stenosis (>70%). Bilateral carotid involvement was diagnosed in 23 (34.3%) patients; 60 (89.5%) had hypertension at the time of admission; 55 (82.1%) also had coronary artery disease; 11 (16.4%) had associated peripheral arterial disease; and in 16 of them, two or three arterial beds were involved (carotid, coronary, limb). Sixteen patients had diabetes (type II, 23.9%), and all but one had some form of dyslipidemia (total cholesterol >200 mg/dL, serum triglycerides >150 mg/dL). Fourteen cases had neutrophilia (PMNs > 7.0 × 10^9^/L), three lymphocytosis (LYMPHs > 3.5 × 10^9^/L) and nine monocytosis (MONOs > 900 × 10^9^/L) (Table 1).

### 3.2. Histological Signs of Complicated Plaque

Most specimens had a large, necrotic lipid core (82%), while 59% showed ulceration, 54% intraplaque hemorrhage and 19% atherothrombosis. Microcalcifications were present in 53%, while macrocalcifications were present in 25% of the samples (Table 1).

### 3.3. Correlation of the CD68+ Infiltrate Grade with Signs of Plaque Complication

Macrophage density in the non-core lesion area was scored as previously described and expressed as weak (score 1) or strong (score 2, 3) infiltrate. In a 2 × 2 contingency analysis, these categories did not show significantly different distributions for ulceration, thrombosis or neovascularization. However, CD68+ scores 2/3 were significantly more associated with intraplaque hemorrhage than score 1 (*p* = 0.003), with 24 cases (60%) in the first group and only 6 patients (23%) in the second group (Table 2).

### 3.4. Comparison of the Arg1-Dominant vs. the iNOS2-Dominant Groups

We used digital morphometry to assess CD68+ macrophages as the main elements of the cellular infiltrate. We also quantified the Arg1+ and iNOS2+ surface area within the “inflammatory hotspot” surrounding the lipid core to determine the dominance of the M2 or M1 subtype. We defined a cut-off value one and divided our cases into Arg1-dominant (n = 55) and iNOS2-dominant (n = 12) cases. The two groups showed a significant difference in the occurrence of atherothrombosis (*p* = 0.046), with the iNOS2+-dominant specimens presenting this complication more frequently (41.7% vs. 14.5%). None of the other plaque characteristics showed a significantly different distribution between the groups. The CD68+ surface was almost the same in both groups (1.38% (Arg-1-dominant) vs. 1.33% (iNOS2-dominant)). The absolute numbers of neutrophils, lymphocytes, monocytes and the neutrophil/lymphocyte ratio were comparable between the groups (Table 3).

### 3.5. Correlation of Neovascularization with Other Histological Signs of Plaque Instability

Perilesional inflammation was associated with plaque neovascularization in only 41 cases. In these cases, neovascularization, observed as the presence of newly formed intraplaque microvessels, was associated with a significantly higher CD31+ surface area than in 26 samples without this phenomenon (1.07 ± 0.14% vs. 0.05 ± 0.05%, *p* < 0.001). Ulceration (n = 40) and hemorrhage-positive plaques (n = 30) had slightly, but not significantly, higher values of CD31+ surface area than their negative counterparts (0.72 ± 0.16% vs. 0.61 ± 0.11%, *p* = 0.607 and 0.69 ± 0.20% vs. 0.66 ± 0.11%, *p* = 0.324). In contrast, in the 13 plaques with atherothrombosis, the CD31-positive area was significantly higher than in the plaques without atherothrombosis (1.02 ± 0.20 vs. 0.61 ± 0.12, *p* = 0.020) (Figure 5).

## 4. Discussion

In this cohort study of endarterectomy specimens obtained from patients with symptomatic stenosis, we have provided a detailed oversight of the inflammatory composite elements that make up the ulcerated hemorrhagic and potentially unstable micro-environment within what are essentially heterogeneous atherosclerotic plaques. It is important to note that we included exclusively those cases from our previous work which did show a mononuclear cell infiltrate around the atheroma core lesion (n = 67). Our specific goal was to further define the relationship between active macrophage infiltration, the neoangiogenic processes, and potential thrombotic capacity within the developing intimal core.

The co-expression of immature irregular and leaky intimal microvessels has previously been linked to the development of ‘soft’ inflammatory and foam cell-loaded plaques susceptible to rupture [7,8]. This is often associated with significant inflammatory infiltrate concomitant with extracellular matrix breakdown and thinning of the protective fibrous cap. Macrophages, as mobile elements sensitive to danger signals and effectors heavily loaded with degradative enzymes and reactive oxygen species, possessing reparative capacities, are by far the most important players in the structural transformations of the atheroma. The modified American Heart Association consensus classification clearly defines the evolutional stages from cap thinning to erosion and rupture, where macrophage accumulation and metalloproteinase secretion have a key role [32]. The whole spectrum of macrophages comprises a variety of phenotypes, among which we find the destructive and pro-inflammatory M1 and the opponent M2 subtype endowed with reparatory functions. There are well-known classification markers of these macrophage subclasses, and several studies have previously analyzed their characteristics and distribution in carotid atheromatous plaques. Stöger et al. investigated the distribution of M1- and M2-type macrophage subpopulations in stable and unstable carotid plaques, post-mortem aortic plaques and early lesions of mixed origin. These authors examined the expression of CD68, iNOS, CD86, HLA-DP/Q/R, dectin-1, the scavenger receptor MARCO and CD163 in plaque shoulders, fibrous caps and adventitial tissue. They found that both M1 and M2 markers were overexpressed in these lesions and described a differential distribution of the two cell subtypes in the rupture-prone regions. M1-associated patterns such as iNOS, MHC II and CD86 were characteristic of foam cells, and the class A scavenger receptor, MARCO (macrophage receptor with collagenous structure), showed low expression, whereas M2 type 2 markers, mannose receptor, dectin-1 and CD163 accumulated in the perivascular adventitial tissue [33]. De Gaetano et al. described the accumulation of macrophages, predominantly of the M1 subtype, and smooth muscle cells in the vessel wall of symptomatic plaques. Symptomatic plaques were associated with greater hemorrhagic activity and signs of fibrosis, necrosis and calcification, whereas M2 markers (MR, CD163 and dectin-1) were more strongly expressed in stable plaque regions [16]. Shaikh et al. studied 32 patients with carotid endarterectomy and 25 with femoral endarterectomy and classified the intraplaque subpopulations with CD68, iNOS, MHCII and SOCS3 (M1-type) versus CD68, dectin-1, CD163 and SOCS-1 (M2-type). Both CD3+ T cells and CD68+ macrophages were more abundant in carotid plaques. Expressions of iNOS, MHC II and SOCS 3 were higher, whereas CD163, SOCS-1 and dectin-1 expressions were lower in carotid samples [34]. Another study confirmed a higher vulnerability of symptomatic carotid plaques and an upregulation of CD68+ and CD11c+ cells, in contrast to the CD163+ infiltrate of asymptomatic atheromatous lesions [35].

CD68 is a type I transmembrane glycoprotein widely used for immunohistochemical identification of macrophages. Some bias has been shown in its associated proteomic profiles; specifically, it is associated with stronger iNOS2 and Arg1 expressions than CD163 [36]. This was one reason why we used a simplified approach to characterize the intra-plaque infiltrate, quantifying iNOS2 and Arg1; another important consideration was that these enzymes define two mutually regulated steps of the same biochemical pathway, arginine degradation.

Here, we found that in those plaques with mononuclear cell invasion, the intensity of the CD68 infiltrate was significantly associated with intraplaque hemorrhage. We have characterized the expression of CD68-positive macrophages within the intimal cores of studied plaques, showing that higher macrophage presence was associated with more evidence of plaque hemorrhage, and more specifically, in these inflammatory ‘hotspots’, where there was evidence of atherothrombosis, M1 macrophages (designated by staining with anti-iNOS-2) were the predominant phenotype [12,13,14]. We also highlighted that increased CD31 expression, as a sign of neovascularization, was characteristic of cases with atherothrombosis. It is important to note that we only calculated the CD31 positivity associated with the neovessels in this comparison. Thirdly, we have shown that higher iNOS2 expression is associated with atherothrombosis, and this result is in line with the results mentioned above. Moreover, our protocol was more focused since we specifically studied the infiltrate of the region surrounding the lipid core (regardless of its size and necrotic nature), excluding the foam cells.

In our cohort, more than half of the 67 histological specimens showed microcalcification, which, in our previous study, proved to be a strong determinant of plaque ulceration [28]. However, in the present cases, all with a mononuclear cell inflammatory infiltrate, none of the calcification patterns showed significant associations with the subtype of the macrophages (Arg1 or iNOS2 dominant).

In general, angiogenesis seen within growing plaques indicates instability and likely hemorrhage. Evidence has shown that endothelial cells of neointimal vessels can originate following activation and trans-migration of vasa vasorum of the adventitia. Hence, strong pro-inflammatory signaling from macrophage-rich microenvironments could be the switch that instigates this [37]. In this study, microvessels from the intima showed heterogeneous size, shape, and patency, some bearing stabilizing smooth muscle cells. The frequency was correlated with atherothrombotic regions.

Several studies have indicated that blocking macrophage activity in vivo could help stabilize and slow down the growth of arterial plaques. For example, Tang et al. used nanoparticles to direct simvastatin to infiltrate regions in ApoE4 mice, successfully blocking macrophage proliferation and slowing plaque growth [38]. Li et al. summarized the importance of M1 macrophages as instigators of plaque progression, suggesting the possibility of their use as a clinical biomarker [39]. Macrophage infiltration is not the exclusive determinant of their accumulation in the atheroma. In the past years, it has turned out that macrophage retention is also a crucial, signal-driven process [40]. As part of patients’ lipid-lowering strategies are inefficient, macrophage dynamic polarization and flux evaluation will become helpful tools for assessing therapeutic efficiency to reach regression. Methodologically, various in vivo imaging techniques have been proposed, like Förster-resonance energy transfer with fluorescent reporters or sodium iodide symporter coupled with 99mTc-single-photon emission computed tomography [40,41]. These results suggest the importance of plaque inflammatory activity monitoring in at-risk patients, and apparel functional, non-invasive, and inexpensive imaging techniques with such a potential [42].

One limitation of this study was that we did not characterize the currently active microvessels using antibodies such as CD105. CD105 proved to be more sensitive than CD31 in staining microvessels in carotid arteries of diabetic patients, possessing a greater affinity for activated endothelial cells and correlated with the grade of stenosis in advanced carotid atherosclerosis [43,44]. Another limitation is that we studied a cohort with a relatively reduced number of subjects. Our study was retrospective and the observations made exclusively on severe carotid stenosis patients with recommendation for carotid endarterectomy. Thus, the results cannot be generalized to other populations with less severe carotid stenosis, but with vulnerable plaques. Further studies should aim to examine in more detail the relationship between actively growing immature vessels, macrophage phenotype and local secretion of matrix destabilizing proteins. Future targeting of arterial ‘hotspots’ encouraging fibrous proliferation might be one therapeutic mechanism to stabilize potentially thrombotic intimal zones [45], while macrophage M1-radiolabelled tracers could support enhanced imaging to identify vulnerable regions at risk of thrombosis [46].

## 5. Conclusions

In this study, we focused on the carotid atherosclerotic plaques with an active perilesional inflammatory infiltrate and correlation with their histological and immunochemistry signs specific for unstable plaques: ulceration, thrombosis, intraplaque hemorrhage, lipid core, calcification and neovascularization.

Our results release the following aspects about inflammatory atherosclerotic plaques of patients with severe carotid stenosis: (1) most of the plaques had a large necrotic lipid core, while ulceration, intraplaque hemorrhage and microcalcifications were present in more than 50% of the cases; (2) massive infiltrate with CD68+ macrophage in lipid-core surrounding lesional area was associated with intraplaque hemorrhage; (3) considering the non-foam cell macrophage population, the M2 subtype of macrophage was the dominant one in our specimens, but the subtle representation of the iNOS2+ M1 subtype in the hotspot region was significantly associated with atherothrombosis; (4) plaque neovascularization identified with CD31+ high areas were correlated with atherothrombosis; (5) absolute neutrophil, lymphocyte, monocyte counts and neutrophil/lymphocyte ratios were comparable between M1- and M2-dominant groups.

Based on our results, we hypothesize that even if we have carotid plaques that create high-grade artery stenosis, their complications, specific to the unstable plaques, may appear and lead to neurological disorders like transient ischemic attack or stroke. Furthermore, multidisciplinary studies are needed to prevent the development of complications from carotid atherosclerotic inflammatory plaques.

## Figures and Tables

**Figure 1 biomedicines-11-03275-f001:**
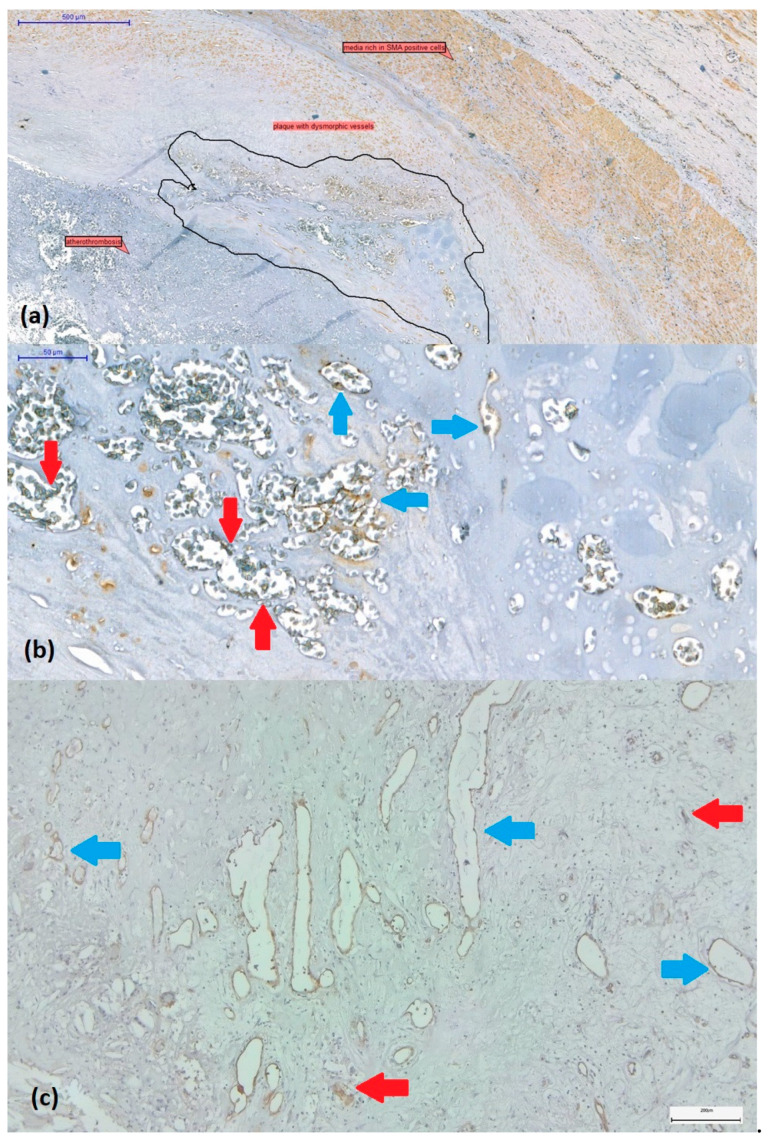
(**a**) SMA-immunostained carotid artery wall fragment affected by atherothrombosis at the level of the plaque with a proliferation of newly formed vessels ranging from microvessels (with reduced/collapsed lumen) to dilated branching vessels with irregular lumen (circled). (**b**) These immature and dysmorphic vessels lack SMA-positive smooth muscle cells (blue arrows) or show discontinuity of SMA-positive immunolabelled coverage (red arrows). (Immunolabel was reported to be positive for endogenous control on media and myofibroblasts within the plaque, visualized by 3,3’-diaminobenzidine chromogen, 10× magnification.) (**c**) Revascularized plaque with small to large, thin-walled, neovascularized vessels covered by CD31-positive endothelium (blue arrows). Immature vascular elements in the form of endothelial cell buds (red arrows) can also be observed (CD31 immunohistochemistry in combination with 3,3’-diaminobenzidine chromogen, original magnification × 4).

**Figure 2 biomedicines-11-03275-f002:**
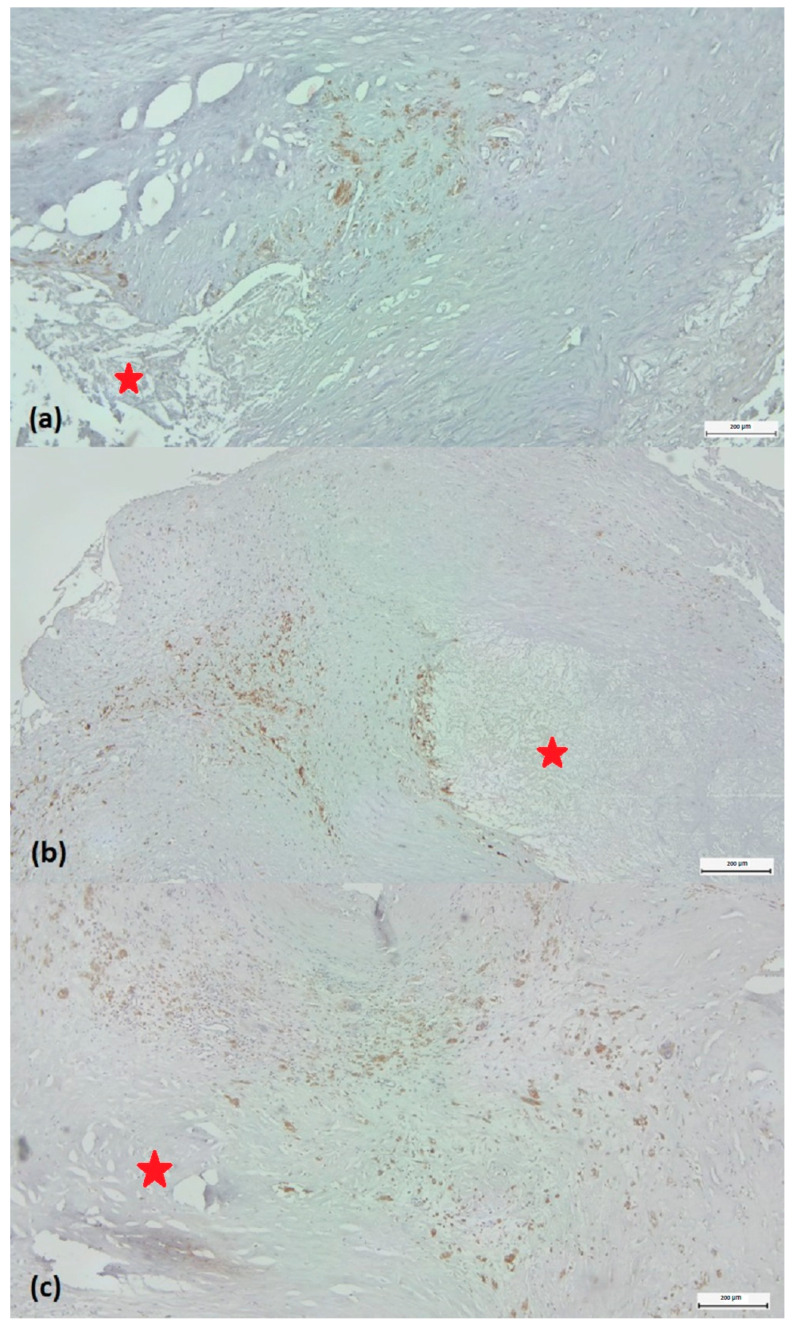
Scoring of the density of CD68 labeled monocytes/macrophages around the plaque. (**a**) Score 1: few positive CD68 cells representing less than 5% of the cell population around the lipid core (marked with a red star). (**b**) Score 2: immunolabelled cells between 5 and 10% of the total perilesional cell pool. Score 3: number of positive cells greater than 10%. (**c**) CD68/3,3’-diaminobenzidine chromogen immunohistochemistry, original magnification × 4.

**Figure 3 biomedicines-11-03275-f003:**
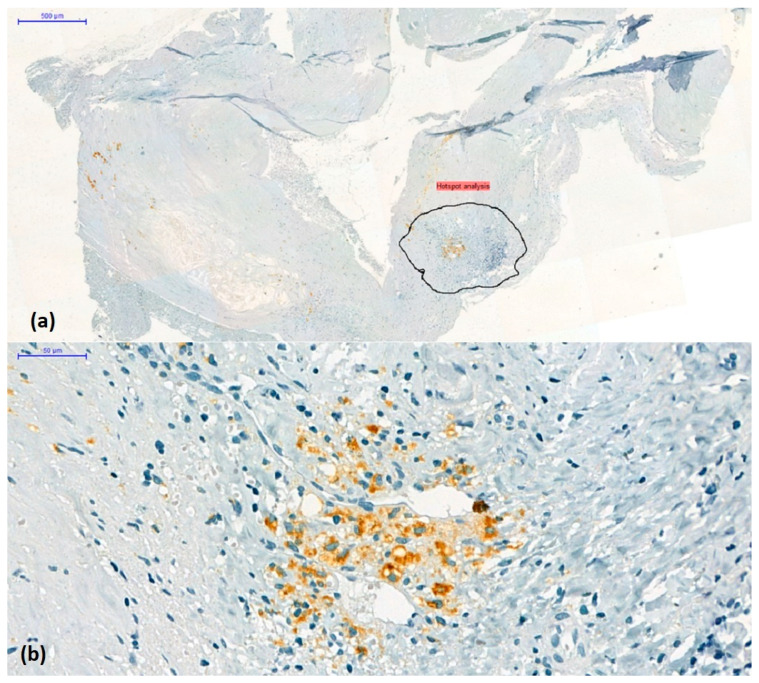
(**a**) Hotspot method: choice and annotation of the most representative regions containing the most immunolabelled elements (original magnification ×2). (**b**) CD68-positive mononuclear cell density in the area selected for digital image analysis (CD68/3,3’-diaminobenzidine chromogen combination, original magnification × 10).

**Figure 4 biomedicines-11-03275-f004:**
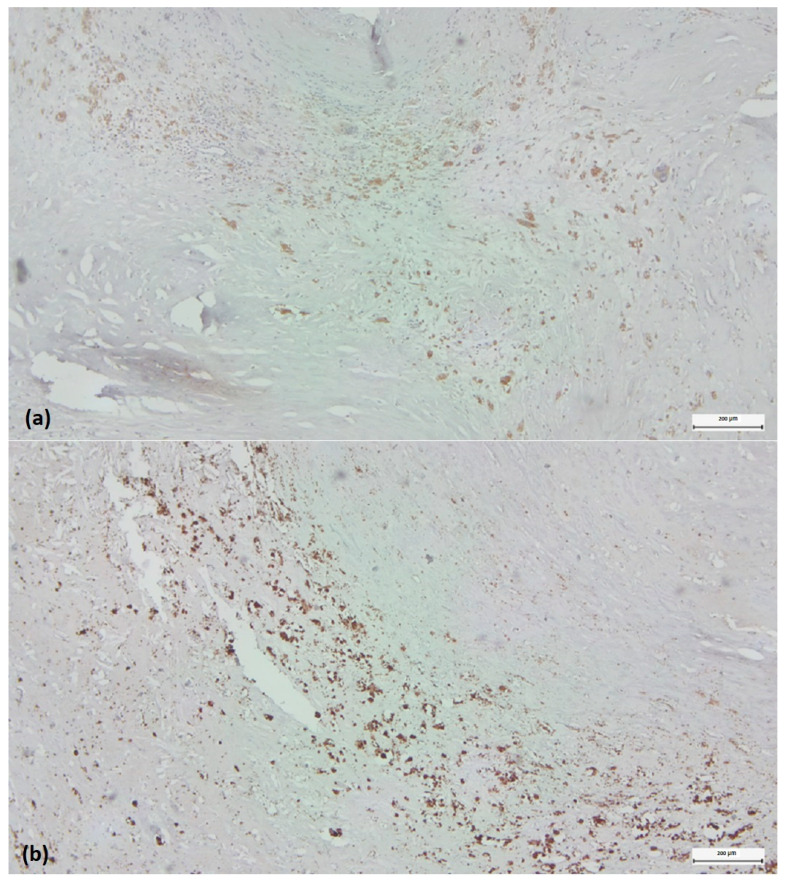
Detection of pro-inflammatory (M1) and anti-inflammatory (M2) monocyte/macrophage subsets in histological regions corresponding to highly reactive cell pools. Arg1+ (**a**) versus iNOS2+ (**b**) cells within the “inflammatory hotspot”. Visualization by immunohistochemistry (3,3′-diaminobenzidine chromogen, original magnification × 4).

**Figure 5 biomedicines-11-03275-f005:**
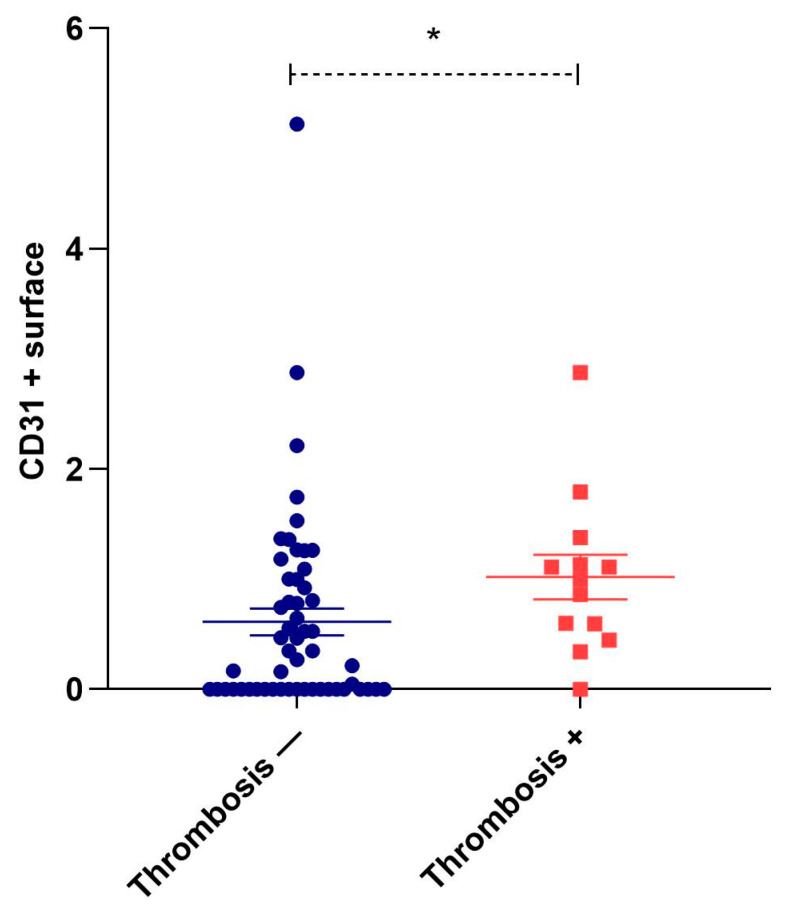
Dot-plot representation of the intraplaque CD31+ surface area in subgroups with and without thrombosis. Values are represented as percentages, mean ± SE. * *p* < 0.05.

**Table 1 biomedicines-11-03275-t001:** Demographic factors, atherosclerotic plaque characteristics, immunohistochemistry parameters, complete blood count parameters.

Demographic Factors	
Age (years)	65.4 ± 1.1
Gender (male/female)	47 (70.1)/20 (29.9)
**Plaque characteristics**	
Ulceration (yes/no)	40 (59.7)/27 (40.3)
Atherothrombosis (yes/no)	13 (19.4)/54 (80.6)
Intraplaque hemorrhage (yes/no)	30 (54.5)/37 (45.5)
Necrotic lipid core (yes/no)	55 (82.1)/12 (17.9)
Microcalcification (yes/no)	36 (53.7)/31 (46.3)
Superficial/deep calcification (yes/no)	25 (37.3)/42 (62.7)
Macrocalcification (yes/no)	17 (25.3)/50 (74.7)
Neovascularization (yes/no)	41 (61.2)/26 (38.8)
**Immunohistochemistry parameters**	**Positive surface area**
CD68+ surface (%)	1.37 ± 0.14
iNOS2 + surface (%)	0.48 ± 0.08
Arg1 + surface (%)	1.79 ± 0.20
CD31 + surface (%)	0.69 ± 0.10
**Complete blood count parameters**	
Neutrophils (10^9^/L)	5.89 ± 0.28
Lymphocytes (10^9^/L)	2.11 ± 0.09
Neutrophil/Lymphocyte ratio	3.38 ± 0.67
Monocytes (10^9^/L)	0.76 ± 0.08
**Medication**	
Anti-hypertensive drugs (yes/no)	65 (97)/2 (3)
Anticoagulants	63 (94)/4 (6)
Anti-aggregants	67 (100)/0 (0)
Statins	65 (97)/2 (3)

Continuous variables are presented as mean ± SE. Binomial variables are presented as absolute numbers, and percentages are given in brackets.

**Table 2 biomedicines-11-03275-t002:** Correlation of the CD68+ infiltrate grade with plaque complication.

	CD68+ Infiltrate Density		
	Score 1	Score 2–3	*p* Value
Ulceration (yes/no)	14/12	26/14	0.318
Atherothrombosis (yes/no)	5/21	8/32	1
Intraplaque hemorrhage(yes/no)	6/20	24/16	0.003
Neovascularization (yes/no)	18/9	23/17	0.609

Incidence is shown as an absolute number. *p* values calculated by Fisher’s exact test.

**Table 3 biomedicines-11-03275-t003:** Correlation of Arg1+ and iNOS2+ with demographic factors, plaque complications and complete blood count characteristics.

Groups	Arg1/iNOS2 ≥ 1 (n = 55)	Arg1/iNOS2 < 1 (n = 12)	*p* Value
**Demographic factors**			
Age (years)	65.5 ± 1.2	64.8 ± 1.9	
Gender (male/female)	36 (65.5)/19 (34.5)	11 (91.6)/1 (8.4)	0.090
**Plaque characteristics**			
Ulceration (yes/no)	33 (60)/22 (40)	7 (58.3)/5 (41.7)	0.990
Atherothrombosis (yes/no)	8 (14.5)/47 (85.5)	5 (41.7)/7 (58.3)	0.046
Intraplaque hemorrhage (yes/no)	23 (41.8)/32 (58.2)	7 (58.3)/5 (41.7)	0.348
Necrotic lipid core (yes/no)	45 (81.8)/10 (18.2)	10 (83.3)/2 (16.6)	1.000
Microcalcification (yes/no)	29 (52.7)/26 (47.3)	7 (58.3)/5 (41.7)	0.760
Superficial/deep calcification(yes/no)	19 (34.5)/36 (65.5)	6 (50)/6 (50)	0.345
Macrocalcification (yes/no)	13 (23.6)/42 (76.4)	4 (33.3)/8 (66.6)	0.482
Neovascularization (yes/no)	31 (56.3)/24 (43.7)	10 (83.3)/2 (16.6)	0.108
CD68 + area (%)	1.38 ± 0.16	1.33 ± 0.35	0.740
CD31 + area (%)	0.68 ± 0.12	0.75 ± 0.23	0.502
**Complete blood count** **parameters**			
Neutrophils (10^9^/L)	5.72 ± 0.29	6.65 ± 0.78	0.226
Lymphocytes (10^9^/L)	2.08 ± 0.09	2.24 ± 0.23	0.491
Neutrophil/Lymphocyte ratio	3.27 ± 0.36	3.38 ± 0.67	0.692
Monocytes (10^9^/L)	0.64 ± 0.03	0.76 ± 0.08	0.245

Continuous variables are presented as mean ± SE. Binomial variables are presented as absolute numbers and percentages in brackets. Comparison of variables with discrete values was performed using Fisher’s exact test (2 × 2 groups) and Pearson χ^2^ test (3 × 2 groups). For continuous variables, the Mann–Whitney U test was used to compare groups. The level of statistical significance was set at *p* = 0.05.

## Data Availability

Data spreadsheets available as Nagy, Előd Ernő (2023), “Data inflammatory carotid endarterectomy specimens_IAB_HE_2023”, Mendeley Data, V1, doi: https://doi.org/10.17632/k48xd5gkgh.1 (accessed on 1 October 2023).

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
