# Peer review of "Intraplaque Neovascularization, CD68+ and iNOS2+ Macrophage Infiltrate Intensity Are Associated with Atherothrombosis and Intraplaque Hemorrhage in Severe Carotid Atherosclerosis"

_biomedicines, 2023, doi:10.3390/biomedicines11123275_

Round 1

Reviewer 1 Report

Comments and Suggestions for Authors

Balmos and colleagues conducted an extensive investigation into the composition of CEA (carotid endarterectomy) plaques and their associated instabilities. They arrived at a set of noteworthy conclusions from their well-constructed and eloquently written study. They found that higher levels of CD68+ were linked to intraplaque hemorrhage, along with the prevalence of iNOS-type M1 macrophages. Additionally, they observed a more frequent association between CD31 expression and atherothrombosis.

To further strengthen the manuscript, the following questions should be addressed:

1.      Intraplaque hemorrhage definition: It would be beneficial for the authors to clarify how they define intraplaque hemorrhage, especially given concerns about the suitability of H/E staining in cases where CEA samples may not be adequately perfused. Consideration of using Glycophorin IHC staining could help enhance the accuracy of this assessment.

2.      Atherothrombosis definition: Similar questions arise regarding the definition of atherothrombosis. The authors may want to specify if they conducted staining for fibrin or platelets to better characterize this critical aspect of plaque instability.

3.      Specific staining for calcification: To provide a more detailed understanding of the plaque composition, the authors should indicate if they used specific staining techniques to delineate calcification within the plaques.

4.      Statin treatment and calcification: It would be insightful to know whether the patients involved in the study were on statin treatments, as some reports suggest that statins may contribute to calcification. Addressing this point would add another layer of context to the findings and their potential clinical implications

Author Response

Dear Reviewer,

Thank you for your careful revision and valuable comments.

Here, we send our point-to-point answers to the issues raised. We also cited the modifications performed in the manuscript, which are shown in Italics.

Reviewer:

Balmos and colleagues conducted an extensive investigation into the composition of CEA (carotid endarterectomy) plaques and their associated instabilities. They arrived at a set of noteworthy conclusions from their well-constructed and eloquently written study. They found that higher levels of CD68+ were linked to intraplaque hemorrhage, along with the prevalence of iNOS-type M1 macrophages. Additionally, they observed a more frequent association between CD31 expression and atherothrombosis.

To further strengthen the manuscript, the following questions should be addressed:

  1. Intraplaque hemorrhage definition: It would be beneficial for the authors to clarify how they define intraplaque hemorrhage, especially given concerns about the suitability of H/E staining in cases where CEA samples may not be adequately perfused. Consideration of using Glycophorin IHC staining could help enhance the accuracy of this assessment.

Authors: we introduced the following definition and specifications in the manuscript:

“Intraplaque hemorrhage was defined by the extravasation and accumulation of blood components and fibrin deposition within the atheromatous plaque clearly visible on H&E stained sections [1,2]. However, for the selection of cases with this modification, we also used CD31-immunostained sections, which visualize both the density of neovascularization and the rupture of the neovascular wall with consecutive intraplaque hemorrhage. CD31 immunolabeling has also helped to characterize atherothrombosis more accurately, indicating a lack of continuity of the intima at the site of clot adhesion to the inner arterial wall.”

  1. Atherothrombosis definition: Similar questions arise regarding the definition of atherothrombosis. The authors may want to specify if they conducted staining for fibrin or platelets to better characterize this critical aspect of plaque instability.

Authors: Plaque erosion and disruption, platelet aggregation, tissue factor release and fibrin formation are the major signs of atherothrombosis [3]. Intimal erosion, plaque disruption and consequent platelet accumulation are detectable with routine hematoxylin-eosin staining [4,5]. Therefore, we did not use specific staining for platelets or fibrin deposition. We introduced the following definition in our manuscript:

“Atherothrombosis was defined as plaque disruption and consequent platelet deposition on the injured vessel wall.”

  1. Specific staining for calcification: To provide a more detailed understanding of the plaque composition, the authors should indicate if they used specific staining techniques to delineate calcification within the plaques.

Authors: the methodological approach, and the classification of calcification patterns has been adopted from previously published works [6,7]. We used hematoxylin-eosin and von Kossa staining to highlight calcium deposition in our specimens. We attach a small representative picture of von Kossa specific staining.

We have completed the Materials and methods section with the following sentence:

“Since the type of calcification in fibrohialinous lesions of atheromatous plaque is difficult to distinguish from H&E staining and can be misleading, we also used von Kossa's special staining to clarify the pattern of calcification, with particular attention to the identification of microcalcification foci.

  1. Statin treatment and calcification: It would be insightful to know whether the patients involved in the study were on statin treatments, as some reports suggest that statins may contribute to calcification. Addressing this point would add another layer of context to the findings and their potential clinical implications

Authors: indeed, statin therapy might have been influence our results. In animal studies pravastatin increased the size of microcalcification deposits [8], and coronary artery calcium scores were increased after prolonged statin therapy [9]. However, almost all, 65 patients in our cohort did benefit of statin therapy, the duration of which was not evaluated. Therefore, we think that we have homogeneity concerning treatment, and this condition could not bias our results, which were not significant for the distribution of calcification patterns.

References

  1. Levy AP, Moreno PR. Intraplaque haemorrhage. Curr Mol Med. 2006 Aug;6(5):479-88. doi: 10.2174/156652406778018626.
  2. Mura M, Della Schiava N, Long A, Chirico EN, Pialoux V, Millon A. Carotid intraplaque haemorrhage: pathogenesis, histological classification, imaging techniques and clinical value. Ann Transl Med. 2020 Oct;8(19):1273. doi: 10.21037/atm-20-1974.
  3. Asada Y, Yamashita A, Sato Y, Hatakeyama K. Pathophysiology of atherothrombosis: Mechanisms of thrombus formation on disrupted atherosclerotic plaques. Pathol Int. 2020 Jun;70(6):309-322. doi: 10.1111/pin.12921
  4. Asada Y, Yamashita A: Pathophysiology of Atherothrombosis — Thrombus Growth, Vascular Thrombogenicity, and Plaque Metabolism. In: Thrombosis, Atherosclerosis and Atherothrombosis - New Insights and Experimental Protocols http://dx.doi.org/10.5772/61769

  1. Kim JS, Lee SG, Oh J, Park S, Park SI, Hong SY, Kim S, Lee SH, Ko YG, Choi D, Hong MK, Jang Y. Development of Advanced Atherosclerotic Plaque by Injection of Inflammatory Proteins in a Rabbit Iliac Artery Model. Yonsei Med J. 2016 Sep;57(5):1095-105. doi: 10.3349/ymj.2016.57.5.1095
  2. Wong, K.K., Thavornpattanapong, P., Cheung, S.C. et al.Effect of calcification on the mechanical stability of plaque based on a three-dimensional carotid bifurcation model. BMC Cardiovasc Disord12, 7 (2012). https://doi.org/10.1186/1471-2261-12-7
  3. Shi X, Gao J, Lv Q, Cai H, Wang F, Ye R, Liu X. Calcification in Atherosclerotic Plaque Vulnerability: Friend or Foe? Front Physiol. 2020 Feb 5;11:56. doi: 10.3389/fphys.2020.00056.
  4. Xian JZ, Lu M, Fong F, Qiao R, Patel NR, Abeydeera D, Iriana S, Demer LL, Tintut Y. Statin Effects on Vascular Calcification: Microarchitectural Changes in Aortic Calcium Deposits in Aged Hyperlipidemic Mice. Arterioscler Thromb Vasc Biol. 2021 Apr;41(4):e185-e192. doi: 10.1161/ATVBAHA.120.315737.
  5. Ngamdu KS, Ghosalkar DS, Chung HE, Christensen JL, Lee C, Butler CA, Ho T, Chu A, Heath JR, Baig M, Wu WC, Choudhary G, Morrison AR. Long-term statin therapy is associated with severe coronary artery calcification. PLoS One. 2023 Jul 27;18(7):e0289111. doi: 10.1371/journal.pone.0289111

Reviewer 2

Reviewer 2 Report

Comments and Suggestions for Authors

The authors examined the density and subtype of macrophages, the degree of neovascularization, and the correlation with plaque instability and complications in carotid endarterectomy specimens from patients with symptomatic stenosis. They measured the expression of different macrophage subtypes (M1 and M2) and vascular markers (CD31 and CD105) in carotid plaques from patients with different degrees of stenosis and symptoms. The authors found that M1 macrophages were more prevalent in the hotspot regions of the plaques and were associated with intraplaque hemorrhage and atherothrombosis. CD105-positive neovessels were also correlated with plaque instability and complications. The authors suggest that the balance between M1 and M2 macrophages and the degree of plaque neovascularization may be useful biomarkers for assessing the risk of stroke and other neurological disorders caused by carotid plaques. They also call for more studies to prevent the development of plaque complications. Specific comments:

1.          The abstract is well-written and summarizes the main objectives, methods, results, and conclusions of the study. However, it could be improved by adding some information about the clinical implications of the findings.

2.          The introduction provides a comprehensive background on the pathophysiology of carotid atherosclerosis and the role of inflammation, macrophages, and neovascularization in plaque instability. However, it could be more concise and focused on the specific research question and hypothesis of the study. Some paragraphs could be shortened or moved to the discussion section.

3.          The materials and methods section describes in sufficient detail. However, some information is missing or unclear, such as:

l   The criteria for defining symptomatic and asymptomatic carotid stenosis and the time interval between the onset of symptoms and the surgery.

l   The rationale for choosing the cut-off value of 1 for dividing the cases into Arg1-dominant and iNOS2-dominant groups.

l   The validation and reproducibility of the digital image analysis method and the hotspot selection.

4.          Please provide the scale bars for Figure 3.

5.          The resolution of Figure 5 is not good enough for publication.

6.          The discussion section interprets the results in the context of the existing literature and highlights the strengths and limitations of the study. However, it could be more structured and balanced, by addressing the following points:

l   The comparison and contrast of the results with those of previous studies, with citations and references to support the arguments.

l   The explanation of the possible mechanisms and pathways underlying the observed associations, with reference to the biological and molecular evidence.

l   The discussion of the clinical relevance and implications of the findings, such as the potential use of macrophage subtypes and neovascularization as biomarkers or therapeutic targets for carotid atherosclerosis.

l   The acknowledgement of the limitations of the study, such as the small sample size, the retrospective and observational design, the selection bias, the lack of functional assays, and the generalizability of the results to other populations and settings.

l   The suggestion of future directions and recommendations for further research, such as the validation of the results in larger and prospective cohorts, the exploration of other macrophage markers and subtypes, the evaluation of the effects of interventions on plaque stability, and the development of novel imaging and diagnostic techniques.

Author Response

Dear Reviewer,

Thank you for your careful revision and valuable comments.

Here, we send our point-to-point answers to the issues raised. We also cited the modifications performed in the manuscript, which are shown in Italics.

Reviewer: The authors examined the density and subtype of macrophages, the degree of neovascularization, and the correlation with plaque instability and complications in carotid endarterectomy specimens from patients with symptomatic stenosis. They measured the expression of different macrophage subtypes (M1 and M2) and vascular markers (CD31 and CD105) in carotid plaques from patients with different degrees of stenosis and symptoms. The authors found that M1 macrophages were more prevalent in the hotspot regions of the plaques and were associated with intraplaque hemorrhage and atherothrombosis. CD105-positive neovessels were also correlated with plaque instability and complications. The authors suggest that the balance between M1 and M2 macrophages and the degree of plaque neovascularization may be useful biomarkers for assessing the risk of stroke and other neurological disorders caused by carotid plaques. They also call for more studies to prevent the development of plaque complications. Specific comments:

  1. The abstract is well-written and summarizes the main objectives, methods, results, and conclusions of the study. However, it could be improved by adding some information about the clinical implications of the findings.

Authors: We reformulated the abstract to be more concise, and added the following phrase:

“Modulating macrophage polarization may be a successful therapeutic approach to prevent plaque destabilization.Please, see the revised version.

  1. The introduction provides a comprehensive background on the pathophysiology of carotid atherosclerosis and the role of inflammation, macrophages, and neovascularization in plaque instability. However, it could be more concise and focused on the specific research question and hypothesis of the study. Some paragraphs could be shortened or moved to the discussion section.

Authors: We revised the Introduction section and eliminated several phrases to simplify its content. Please, see the revise Introduction.

  1. The Materials and methods section describes in sufficient detail. However, some information is missing or unclear, such as:

l   The criteria for defining symptomatic and asymptomatic carotid stenosis and the time interval between the onset of symptoms and the surgery.

Authors: We used the European Society for Vascular Surgery (ESVS) 2023 clinical practice guidelines to define symptomatic and asymptomatic carotid stenosis. Thus, symptomatic patients with severe carotid artery stenosis were considered those who presented higher cortical dysfunction (aphasia, dysgraphia, apraxia, visual field deficits), amaurosis fugax (transient monocular blindness blurring), weakness and/or sensory impairment of face/arm/leg, dysarthria or limb shaking.

We completed our statement in Materials and methods as follows:

“The carotid plaque specimens whose morphological characteristics are processed in this study were obtained by endarterectomy from 119 patients diagnosed with symptomatic carotid artery (CA) stenosis (according to the European Society for Vascular Surgery, 2023 definitions) hospitalized between 2020 and 2022 January at the Vascular Surgery Clinic - County Emergency Clinical Hospital and the Cardiovascular Surgery Clinic - Cardiovascular Disease and Transplant Emergency Institute of Târgu Mureș (Romania).

l   The rationale for choosing the cut-off value of 1 for dividing the cases into Arg1-dominant and iNOS2-dominant groups

Authors: The Arg1/iNOS2 ratio in our cohort had a mean ± SE of 88.0 ± 47.5, and median (range) of 5.3 (1.3-13.3). These results show the shift to higher Arg1 expressions, and we tried several forms of categorization for the Arg1/iNOS2 ratio, like tertiles, and ratios wit cut-off=4 and cut-off=1. The latter classification gave the most relevant results, and, mathematically was reasonable to divide the cohort on the base of the predominant enzyme.

l   The validation and reproducibility of the digital image analysis method and the hotspot selection.

Authors: We have opted for digital morphometry of the foci acquired on TIFF format saved images, using the hot spot method, as it gives a high degree of reproducibility to the results obtained. The immunohistochemical DAB staining greatly enhances the possibility of colorimetric segmentation (using HSB-hue, saturation, brightness), after which these images are ready for fully automated analysis, thus excluding the examiner's subjectivity (Ferreira, T., Rasband, W., 2012. ImageJ User Guide. Image processing and analysis in Java. National Institutes of Health, http://rsb.info.nih.gov/ij.)

As validation parameters, we used “repeatability” and “reproducibility”: colorimetric segmentation and analysis of positive signals of hot spot foci were performed twice and compared. At the same time „detection limit” (the lowest value that can be detected) was studied for each immunolabel studied (CD68, iNOS2 and Arg1) and for foci with low cellularity (less than 5 cells).

  1. Please provide the scale bars for Figure 3.

Authors: Figure 3a and 3b both have scalebars (500μm and 50μm).

  1. The resolution of Figure 5 is not good enough for publication.

Authors: We provided the revised Figure 5 in three different formats: .pdf, .tiff and .jpg with 300 dpi resolution, which fits the technical requirements of the journal. Hopefully, the technical editors will use in the final form of the manuscript the best variant.

  1. The discussion section interprets the results in the context of the existing literature and highlights the strengths and limitations of the study. However, it could be more structured and balanced, by addressing the following points:

l   The comparison and contrast of the results with those of previous studies, with citations and references to support the arguments.

l   The explanation of the possible mechanisms and pathways underlying the observed associations, with reference to the biological and molecular evidence.

l   The discussion of the clinical relevance and implications of the findings, such as the potential use of macrophage subtypes and neovascularization as biomarkers or therapeutic targets for carotid atherosclerosis.

l   The acknowledgement of the limitations of the study, such as the small sample size, the retrospective and observational design, the selection bias, the lack of functional assays, and the generalizability of the results to other populations and settings.

 l   The suggestion of future directions and recommendations for further research, such as the validation of the results in larger and prospective cohorts, the exploration of other macrophage markers and subtypes, the evaluation of the effects of interventions on plaque stability, and the development of novel imaging and diagnostic techniques.

Authors: thank you for these useful suggestions. We have revised and re-organized the Discussion section, which now is structured according to the following points:

  1. the general role of macrophages in plaque formation and progression,
  2. functional phenotypic subtypes and their markers,
  3. previous observations from the literature concerning the presence of these subtypes in atherosclerotic plaques,
  4. our results with special emphasize on relationships between CD68 and hemorrhage, iNOS2, CD31 and atherothrombosis
  5. the potential role and future directions to assess in vivo the macrophage infiltrate,
  6. potential therapeutic implications and the study limitations.

Please, see the revised Discussion section.

Reviewer 3 Report

Comments and Suggestions for Authors

This is an immunohistochemical study in endarterectomized carotid plaques. The main findings are that macrophage infiltration correlated with intraplaque hemorrhage, and the presence of low Arg1/iNOS2+ macrophages and CD31+ was associated with atherothrombosis.

There are some results needing further discussion, mainly:

-         - Did the authors expect more macrophage infiltration (CD68+) in iNOS+?

-         - Was CD68+ associated with CD31+?

-    -  In Arg1/iNOS<1 the presence of atherothrombosis was 7/5 that is not so clear to conclude a direct relationship.

-     -    Do the authors have information about imaging techniques (US, angio-CT) of the carotid plaques of the patients? It would have been very interesting to associate the immunohistochemical parameters with imaging analysis.

Other comments:

-          The medical treatment of the patients (lipid-lowering or others) should be indicated

-          A Figure showing the % of plaques with hemorrhage according to the CD68+ infiltration score could be added to the manuscript.

-          The text indicates that Figure 5 shows CD31+ in plaques with and without atherothrombosis, but in the Legend (and Figure) the groups are Arg1 and iNOS2 dominant groups.

Author Response

Dear Reviewer,

Thank you for your careful revision and valuable comments.

Here, we send our point-to-point answers to the issues raised.

This is an immunohistochemical study in endarterectomized carotid plaques. The main findings are that macrophage infiltration correlated with intraplaque hemorrhage, and the presence of low Arg1/iNOS2+ macrophages and CD31+ was associated with atherothrombosis.

There are some results needing further discussion, mainly:

-         - Did the authors expect more macrophage infiltration (CD68+) in iNOS+?

Authors: this was a cohort study, and we did not compare the iNOS expression with other groups. As in the initial study we enrolled 119 patients (published recently as https://doi.org/10.3390/biomedicines11030881), which were divided according to the presence or absence of the inflammatory infiltrate, the subgroup with no mononuclear cell invasion was inadequate for iNOS and Arg staining. In a previous experimental stroke study, in the rat ischemic core region we found approximately 0.8%, and in the penumbra zone 1.3% iNOS+ surface (https://www.jni-journal.com/article/S0165-5728(18)30225-X/pdf), both regions affected by the immediate post-stroke inflammation, the result obtained here is not surprising.

-         Was CD68+ associated with CD31+?

Authors: yes, we analyzed:

  1. the correlation between the CD68+ and CD31+ areas, which gave: R=-0.067 and p=0.589.
  2. the distribution of CD68 expression categories (increasing numbers 1 to 3 representing increasing expression grade) in the CD31- and CD31+ groups, which is shown in the below contingency table:

CD31-

CD31+

CD68 cat

1

9

17

2

10

11

3

7

12

As it can be seen, no significant relationship could be highlighted.

Reviewer:  In Arg1/iNOS<1 the presence of atherothrombosis was 7/5 that is not so clear to conclude a direct relationship.

Authors: according to our results, 8/55 cases (14.5%) in the Arg1/iNOS>1 subgroup showed atherothrombosis, which is significantly lower, than the proportion observed in the Arg1/iNOS<1 subgroup, 5/12 (41.7%). This does not mean that iNOS expression equals atherothrombosis, however, the probability is higher when iNOS predominates over Arg1.

Reviewer: Do the authors have information about imaging techniques (US, angio-CT) of the carotid plaques of the patients? It would have been very interesting to associate the immunohistochemical parameters with imaging analysis.

Authors: indeed, such a comparison were of great interest. Unfortunately, we weren’t in the possession of intravascular ultrasound, angio-CT or SPECT data to compare with the immunohistochemistry results. It is one of our goals to extend our activity in this direction, and to perform future studies.

Other comments:

Reviewer: The medical treatment of the patients (lipid-lowering or others) should be indicated

Authors: we introduced a sub-section in Table 1, indicating the incidence of anti-hypertensive, anti-coagulant, anti-aggregant drugs and statin administration in our cohort.

Reviewer: A Figure showing the % of plaques with hemorrhage according to the CD68+ infiltration score could be added to the manuscript.

Authors: Thank you for this suggestion. We show the distribution of hemorrhage positive and negative plaques in the CD68 1 vs. 2/3 categories in Table 2, and we think a simple column graph or a variant would not bring additional value to the manuscript.

Reviewer: The text indicates that Figure 5 shows CD31+ in plaques with and without atherothrombosis, but in the Legend (and Figure) the groups are Arg1 and iNOS2 dominant groups.

Authors: thank you for this observation, we corrected Figure 5 allocating the appropriate values and categorization.

Round 2

Reviewer 3 Report

Comments and Suggestions for Authors

The authors have answered my concerns. In my opinion, the manuscript has been improved.